# Learning to Cycle: Are Physical Activity and Birth Order Related to the Age of Learning How to Ride a Bicycle?

**DOI:** 10.3390/children8060487

**Published:** 2021-06-08

**Authors:** Cristiana Mercê, Marco Branco, David Catela, Frederico Lopes, Luis Paulo Rodrigues, Rita Cordovil

**Affiliations:** 1Centro Interdisciplinar de Estudo da Performance Humana, CIPER, Faculdade do Motricidade Humana, Universidade de Lisboa, 1499-002 Cruz-Quebrada, Portugal; marcobranco@esdrm.ipsantarem.pt (M.B.); cordovil.rita@gmail.com (R.C.); 2Escola Superior de Desporto de Rio Maior, Instituto Politécnico de Santarém, 2040-413 Rio Maior, Portugal; catela@esdrm.ipsantarem.pt; 3Motor Behavior, CIEQV, Instituto Politécnico de Santarém Branch, Complexo Andaluz, 2001-904 Santarém, Portugal; 4Faculdade de Motricidade Humana, Universidade de Lisboa, 1499-002 Cruz-Quebrada, Portugal; fred.lopes3@gmail.com; 5Escola Superior de Desporto e Lazer, Instituto Politécnico de Viana do Castelo, 4960-320 Melgaço, Portugal; lprodrigues@esdl.ipvc.pt; 6Research Center in Sports Sciences Health Sciences and Human Development, CIDESD, 5000-801 Vila Real, Portugal

**Keywords:** learning, bicycle, child, birth order, survey

## Abstract

The present article aimed to verify whether the age at which children learn to ride a bicycle is related to their physical activity or birth order. Data were collected from an online structured survey between November 2019 and June 2020. A total of 8614 responses were obtained from 22 countries. The results reveal significant differences in learning age depending on the frequency of physical activity (F(5, 7235) = 35.12, *p* < 0.001, ηp^2^ = 0.24). People who engaged in physical activity less than twice a month learned to cycle later (M = 7.5 ± 5.3 years) than people who engaged in physical activity on a daily basis (M = 5.7 ± 2.2 years) (*p* < 0.001). There were also significant differences in learning age according to birth order (F(2, 3008) = 7.31, *p* = 0.00, ηp^2^ = 0.005). Only children had the highest learning age (M = 5.5 ± 2.4 years), whereas those who were born last had the lowest, (M = 5.1 ± 1.9 years) (*p* = 0.013). Creating opportunities for children to be engaged in play and physical activity and social modulation through their older siblings seem to be key conditions to encourage children to learn how to ride a bicycle from a young age and to foster their motor development.

## 1. Introduction

The concept of physical activity (PA) refers to any bodily movement produced by a muscle’s contraction that substantial increases the energy expenditure above baseline [1], including riding a bicycle. All movement, including getting from one place to another or actively playing with friends during leisure time, or movement that requires significant energy expenditure in a person’s work, is also considered physical activity. Several conceptual models have studied and explored the relationship between the practice of physical activity, motor competence and the health promotion. The World Health Organization [2] recognizes that young children should have opportunities to participate in a range of developmentally appropriate play-based physical activities, which will help them to develop motor competence [3], social and emotional skills [4], and health [5]. In fact, the fundamental role of PA in children’s development is widely recognized [2,6,7].

According to Stodden’s model [8], good levels of motor competence have a key role in promoting healthy trajectories of life concerning PA and weight management. Therefore, motor competence is considered to be a primary mechanism that promotes engagement in PA. Recently, Hulteen and collaborators [9] presented a new conceptual model for PA across lifespan. This model proposes the use of the term “foundational movement skills” instead of “fundamental movement skills”, arguing that foundational movement consists of movement patterns reflecting a broad range of movements that directly or indirectly have an impact on the individual’s capability to be physically active. These movements can be developed to enhance participation in PA and to promote health throughout lifespan. The model argues that these skills should be viewed through a social, cultural and geographic filter. This assumption reinforces the idea that foundational skills are not entirely pre-determined and could vary between different contexts. Activities such as swimming, riding a bicycle or doing push-ups or squats are now considered to be foundational skills, in which children should develop motor competence in order to become more physically active during their lifespan. Ultimately, the model recognizes that the individual’s specific attributes, such as physical characteristics, including weight status or cardiorespiratory fitness, and psychological constructs, such as self-efficacy or perceived competence, also affect the development of these skills and, consequently, the participation in PA across lifespan.

This new model [9] provides a broad view of motor development and its relationship with the promotion of PA and health, highlighting the importance of the new concept of foundational skills. Learning how to ride a bicycle is recognized as one of the foundational movement skills [9,10], and it is also an important motor milestone for children [11]. Cycling is a lifelong skill used for several purposes—as a mode of transportation, in sports, or simply for recreation [10]. Riding a bicycle is a complex skill that allows for fun moments with peers and family [12], promotes greater exploration of the environment and independent mobility in children [13], provides several benefits to physical health, including improvements in cardiorespiratory condition and body composition [14], and to mental health, with the development of emotional and social skills [12,15,16]. These benefits continue throughout life as long as the child, the teenager or the adult continues to cycle; e.g., children who begin to cycle earlier are more likely to have a healthy weight in later school years [17].

The idea that cycle could be a factor that triggers and further promotes physical activity engagement throughout life [9], is corroborated by some intervention studies, namely with children with disabilities, which identified that learning to cycle made children less fearful and more motivated to try other physical and sports activities [18,19]. Children who learned how to cycle spent less time participating in sedentary behaviors, and more time participating in moderate to vigorous physical activity time when compared to control group children [19,20]. In addition, they had better body composition with higher leg strength, and less body fat percentage than children who did not know how to cycle. This results led Hauck et al. [20] to suggest that learning how to ride a bicycle could disrupt the cycle of consistent unhealthy weight gain over time in children with disabilities, which is in line with Hulteen’s suggestion of considering cycling as a foundation movement that promotes PA [9]. However, the relationship between cycling and PA might be bidirectional. Children who engage in more PA are probably also more likely to try cycling, and to learn how to ride a bicycle at an earlier age, than more sedentary children.

Having siblings is another factor that can influence children’s participation in PA and their motor development. However, there is no consensus regarding the effect of having an older sibling in the literature. Some authors claim that older brothers or sisters negatively influence younger siblings’ development, arguing that having siblings implies dividing parental attention, affecting communication opportunities and contributing to a delay in language development [21]. On the other hand, it has been argued that older brothers or sisters positively influence younger siblings’ motor development. Due to social learning, young children tend to observe and imitate older children who are meaningful to them, such as friends or siblings [22]. The social modeling involved in learning a motor skill is also an important aspect of this process. In this sense, learning how to cycle may become a social activity through which siblings create opportunities (i.e., affordances) to play. Having the chance to play with siblings improves cognitive, social and emotional development [23]. In this way, the motor development associated with learning how to ride a bicycle also entails a significant gain for the child in terms of fostering other developmental areas. Although it is relatively consensual that older siblings influence the motor development of the younger ones, the specific characteristics of the family probably also determine the type and magnitude of this influence [24,25].

Considering that riding a bicycle is a foundational skill and an important motor milestone for children, and taking also into account that motor development is influenced by several individual [9] and environmental factors [26], the present study aimed to verify whether the age of learning to ride a bicycle is related with the child’s frequency of physical activity and/or birth order. It has been hypothesized that more physically active children learn to cycle earlier; that younger siblings learn earlier than older ones; and that older ones, in turn, learn earlier than only children.

## 2. Materials and Methods

The present study is part of the international project L2Cycle (Learning to Cycle), which aims to assess different aspects related to the process of learning how to cycle in different countries (e.g., learning age, socio and demographic aspects, type of bicycles used, or who taught the person to cycle). For this purpose, a survey was created on LimeSurvey, hosted by the Faculty of Human Kinetics (University of Lisbon, Lisbon, Portugal), and approved by its ethics committee.

An initial version of the survey was developed by four motor development experts and was tested online on 485 participants, with a sub-sample of 30 participants additionally asked about their comprehension of the survey. Some adjustments were made (e.g., clarifying that the age of learning how to ride should address independent cycling without the help of training wheels or parents). At a second stage, the survey was examined and discussed with five other international experts who provided further suggestions (e.g., adding questions regarding different seasons of the year). Finally, the survey was translated into different languages, now available in 10 languages (Portuguese—from Portugal and Brazil—English, German, Croatian, Finnish, French, Dutch, Italian, Japanese, and Spanish).

The survey has 3 sections: (1) “About you”, questions about the participant’s (adult) personal experience of learning to cycle and demographic data; (2) “About your oldest child” (only if the participant is a mother or father), the same questions as in the previous section, but regarding the participant’s oldest child; (3) “About your youngest child” (only if participant has more than one son/daughter), the same questions as in the previous section, but regarding the participant’s youngest child.

The questions of the survey regarding physical activity were as follows: “When you (your child) learned to ride a bicycle, how often did you (he/she) practice sports, outdoor play, or physical activity?” For this study, six frequencies of PA practice were considered: (1) less than twice a month, (2) twice a month, (3) once a week, (4) two or three times a week, (5) four to six times a week, (6) daily.

The variable birth order had three categories: older, younger or only child. The birth order of the adults was not questioned, and for this reason it was not considered for this study.

The survey was publicized through the social media (Facebook, Instagram and Twitter), and by email. In addition, partnerships with cycling federations, kids and parent’s magazines and non-profit cycling organizations were established in different countries for dissemination on their websites and paper magazines. Data for this study were collected between 22 November 2019 and 8 June 2020. 

Descriptive data analysis was performed to characterize the sample. One-way ANOVAs were used to determine the effects of the frequency of physical activity and birth order on learning age for cycling. Post hoc Scheffé tests were conducted when needed. The level of significance was set at *p* = 0.05.

## 3. Results

There were 8614 responses to this survey. Those responses referred to 4637 adults (self-response) and 3977 children (parental responses). Participants’ mean age was 29.11 years (SD = 17.7), 4975 were male, 3595 were female and 44 preferred not to disclose the sex. Data came from 22 countries: Portugal (2386), Brazil (1556), Italy (1484), Finland (991), United Kingdom (769), Mexico (463), Belgium (438), Croatia (364), Germany (63), Spain (39), USA (21), France (11), Canada (9), Norway (5), Austria (4), Japan (3), United Arab Emirates (2), Bosnia (2), New Zealand (1), Cape Verde (1), Cayman Islands (1), and Taiwan (1).

There was a significant difference in the learning age for cycling depending on children’s frequency of physical activity practice (F (5, 7235) = 35.12, *p* < 0.001, ηp^2^ = 0.24) (Figure 1). Children who practiced physical activity less than twice a month (2×/month) learned significantly later than those who practiced two to three times a week (2–3×/wk), four to six times a week (4–6×/wk) and daily (all *p* < 0.001). Those who practiced 2×/month also learn later than those who practiced 4–6×/wk (*p* = 0.009), and daily (*p* = 0.001). Children who practiced once a week learn later than those who practiced 2–3×/wk (*p* = 0.003), 4–6×/wk (*p* < 0.001) and daily (*p* < 0.001). There was no difference in learning age between children who practiced 4–6×/wk and daily.

A significant difference in learning age was found according to birth order (F (2, 3008) = 7.31; *p* = 0.001, ηp^2^ = 0.005), (Figure 2). Younger children learned earlier than older children (*p* = 0.004) and only children (*p* = 0.013). No significant differences were found between the learning age of older children and only children (*p* = 0.821).

## 4. Discussion

Hulteen’s model [9] highlights the important role of foundational movement skills, such as cycling, to promote and maintain healthy PA trajectories throughout lifespan. In the present study, the causality effect between cycling and PA practice was not possible to address, but a relation between PA and the foundational skill of riding a bicycle was confirmed in the early stages of development. The greater the frequency of PA, the lower the age for learning how to cycle. Children who practiced physical activity more than three times a week learned earlier than all the others, proving the first hypothesis that more physically active children learn to cycle earlier. These results have the same pattern when analyzing the data according to geographical variables (Southern Europe, Northern and Western Europe, and Latin America, all *p* < 0.001). It seems that the relationship between learning to cycle and PA could be bidirectional. In this way, learning to cycle would promote future PA [18,19], and practicing PA would lead to an earlier learning onset age regarding cycling, as we have seen in this study. During childhood, practicing PA, usually through active play, is important for the child to explore and increase his/her motor repertoire, and to develop balance and coordination [6]. When learning how to ride a bicycle, the child should manage and coordinate his/her body with the bicycle, while simultaneously pedaling and balancing. Therefore, coordination and balance are fundamental aspects for cycling. Some authors even claim that balance acquisition is the biggest challenge for cycling [27,28]. Most likely, children who practice PA more frequently have a better chance of developing the necessary skills to learn to cycle, which ultimately leads them to learning at a younger age than children who are more sedentary.

Practicing PA also improves the child’s cardiorespiratory condition and muscular fitness [5]. Some previous studies with children with disabilities pointed to leg strength as a conditioning factor for learning how to cycle [19,29]. Children with lower leg strength developed muscular fatigue more quickly and tended to stop pedaling and training more easily, compromising and/or delaying their cycling acquisition. In typical developing children, leg strength may not be such a conditioning factor in the learning process; however, given that cycling is an activity that requires some cardio and muscular fitness, fitter children would probably learn to cycle more easily. In addition, doing PA also improves phycological attributes, such as perceived competence [30], which in turn tend to increase the engagement in physical activities [31]. Hence, the positive relationship between children’s frequency of practicing physical activity and the age that they learn to cycle is probably influenced by different physical and psychological attributes, as mentioned in Hulteen’s model [9], such as the levels of balance and coordination [6], the cardiorespiratory and muscular fitness [5], and the perceived competence [30]. Another possible cause that might explain the positive relationship between children’s PA and their learning age for independent cycling is that children who practice more PA might have earlier opportunities to practice, because their parents value PA and might give them a bicycle earlier.

The influence of the family, especially of the parents, in the process of learning to cycle has already been approached. Studies indicated that having parents who value cycling and promote its practice leads children to learn how to cycle earlier [18,19]. However, as far as we know, this is the first study to explore the sibling’s influence in the process of learning how to cycle. The results confirm the second hypothesis raised, that younger children learn how to ride a bicycle earlier than older and only children. When considering geographical variables, there were differences between the three regions. In Southern Europe, the younger children learned significantly earlier than the older children (F (2, 1887) = 3.50, *p* = 0.030, ηp^2^ = 0.004), whereas, in Northern Europe (F (2, 678) = 4.40, *p* = 0.013, ηp^2^ = 0.13), younger children learned significantly earlier than only children. There were no significant differences in Latin America (F (2, 431) = 2.67, *p* = 0.71, ηp^2^ = 0.012). Perhaps younger siblings benefit from watching the older ones and even from their help in some cases [22,26]. Other studies, not specifically focused on learning how to cycle, showed that having siblings influences sports participation, and siblings have been suggested to play a key role in sports expertise development [32]. Riding a bicycle is usually an active pleasurable activity to do with younger siblings, increasing their cycling skills and, consequently, accelerating their independent cycle acquisition and expertise. Additionally, siblings’ interactions through play also promote children’s motor and physical development [23], which ultimately might contribute to an earlier acquisition of cycling. In fact, children from 6 to 15 years of age with siblings presented significantly better physical fitness than only children, independent of sex or somatic status [33], while at the preschool age, only children showed lower motor competence than children living with other siblings [34]. Finally, another possible reason for younger siblings to learn how to cycle earlier might be the simple fact that they are more likely to benefit from having an available bicycle to explore and play earlier (i.e., their sibling’s bicycle). Similarly, the fact that the child has someone to copy or imitate was pointed out as a determining factor for the acquisition of the task, which needs to be learned [34,35]. Clearfield et al. [34], in a socialization study focusing on the transition between crawling and walking, found that as infants evolve to a new form of locomotion, they progress from passive to active participants in their social environment, moving from observers to agents of social interaction. Although this was not the focus of our study, and considering our results, this phenomenon may have been the catalyst for the need to learn to ride a bicycle earlier, especially when their social peers (in this case the family) are already doing so. From the perspective of the observing child, cycling can be interpreted as a form of social exploration, as a way to keep up with parents, siblings and/or other children, or as a form of independent exploration of the environment, similar to children who move from crawling to walking, whose visual horizon is broadened.

The third hypothesis, predicting that older siblings would learn to cycle earlier than only children, was not confirmed. Some studies indicate that there are reciprocal effects of sibling relationships on motor development [25]. The idea is that by playing together, both siblings improve motor development, increasing their participation in sports [32]. Cycling with a brother or sister is, probably, more fun than cycling alone. So, having a sibling to cycle with can lead to greater practice and enjoyment, which can promote both children’s learning. The sibling’s interactions could be an influencing and catalyzing factor for learning how to cycle. However, the data from the current study do not support the idea that siblings’ interactions promote both siblings’ motor achievements when it comes to cycling. The analysis of our data suggests that only children are a quite heterogeneous group regarding their cycling learning age; some only children might have benefited from the greater availability of their parents to teach them how to cycle, which might have compensated for the fact that there were no siblings to play with. The interaction between these factors and the relationship between learning age and the number of siblings should be investigated in future studies.

The present research findings reinforce the sociocultural nature of motor development, more specifically of the age from which a child learns how to cycle independently. Such a process is affected by factors, resources, properties, dispositions and constraints made available by the socioecological niche of children’s and families’ lives. Future studies should therefore consider the theoretical perspectives of motor development and task performance as a biosocial process as suggested by authors such as Bronfenbrenner and Morris [36] and Newell [37].

In conclusion, the amount of physical activity that children do is related with their learning age for cycling. Children whose parents report partaking in daily physical activity learn, on average, 1.8 years earlier than those whose parents report exercising less than twice a month. It is possible that physical activity affords better balance, coordination, muscular fitness, and perceived competence, accelerating the learning age for independent cycling. 

Younger siblings learn earlier than older siblings and only children. The younger siblings might benefit from having an available bicycle earlier, and probably also learn by imitation and interaction with their older siblings.

The fact that the amount of physical activity and birth order are related to the learning age for cycling emphasizes the importance of context constraints in motor development during early childhood.

## Figures and Tables

**Figure 1 children-08-00487-f001:**
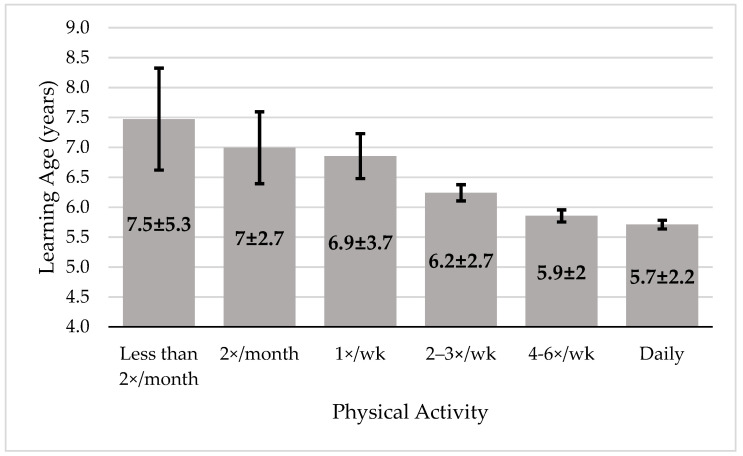
Mean age and standard deviation to learn how to cycle by the frequency of physical activity practice (error bars represent 95% CI).

**Figure 2 children-08-00487-f002:**
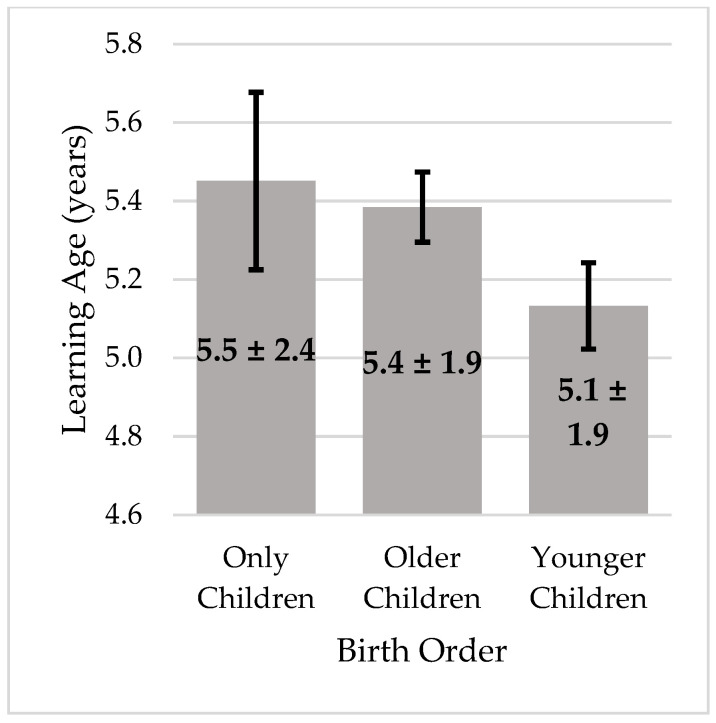
Mean age and standard deviation to learn how to cycle according to the birth order (error bars represent 95% CI).

## Data Availability

The authors have the data available for the journal if it is required.

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
