# Peer review of "Learning to Cycle: Are Physical Activity and Birth Order Related to the Age of Learning How to Ride a Bicycle?"

_children, 2021, doi:10.3390/children8060487_

Round 1

Reviewer 1 Report

I did not see the point of the discussion section as something separated from the conclusions. Also, many of the issues discussed there were not clearly related to the results so I would recommend some trimming.

Author Response

Response to the reviewer 1 MS children-1221034

Reviewer 1

I did not see the point of the discussion section as something separated from the conclusions. Also, many of the issues discussed there were not clearly related to the results so I would recommend some trimming.

Thank you for your suggestions. We decided to merge the discussion and conclusions sections. Regarding the trimming, we were asked by the editor to extend the manuscript, so we added some information in the discussion section. However, we tried to highlight the relationship between the results and the discussion.

Reviewer 2 Report

Dear Authors, This work explores some very interesting and socially important questions around cycling, certainly for Europe. I have a couple of general questions that I think will increase the value of the work further.

1) I personally live in The Netherlands, where roughly every child cycles by the time they turn 5 (go to school). Is there a reason for not taking the Netherlands (or Denmark) in this study? Certainly considering the questionnaire was translated to Dutch! (or was that purely for the flemish part of Belgium?) 

2) And on a similar note: [9] recognises that the foundational movement skills are dependant on the context (ie location). looking at the responses. Roughly 25% of responses are from Latin America, 50% are from southern Europe and 25% from Northern Europe. Are the trends similar with in these regions? Or are they different?

3) Similarly in Northern Europe where there is a significant difference in what can be done outside during winter and summer (especially in Finland), did you take the time of year that the children were born into account? The time of year at which it is possible to learn to ride a bicycle could possibly be just as important as the factors you have investigated. For example learning on ice/snow might be a big cause for learning to cycle half a year later... 

4) I do not understand the numbers mentioned in the bars of figure 1 and 2. for example in figure one in the most left bar (less than 2x/month) it states 7.5+-5.3 . does this mean that the mean age was 7.5 with a standard deviation of 5.3 (ie 2.2years minimum to 12.8 maximum?) then the most right bar (Daily) comes to 3.5 minimum. ie significantly older than the first group... 

5) the following are a couple of details: 

line 129: states 10 languages, it is not clear if there are 2 "Portuguese" languages or just one, as Portugal and Brazil are mentioned, and another 9 languages..  

line 130: Finish = Finnish

line 230: thought = through ? 

Author Response

Response to the reviewer 2 MS children-1221034

Reviewer 2

Dear Authors, This work explores some very interesting and socially important questions around cycling, certainly for Europe. I have a couple of general questions that I think will increase the value of the work further.

Thank you for your nice comment.

1) I personally live in The Netherlands, where roughly every child cycles by the time they turn 5 (go to school). Is there a reason for not taking the Netherlands (or Denmark) in this study? Certainly considering the questionnaire was translated to Dutch! (or was that purely for the flemish part of Belgium?)

Thank you for your comment. In fact, this study is part of an international project and we asked the Netherlands to join from the beginning. However, due to ethics and organization procedures, we only started getting data from the Netherlands after we finished extracting the data for this article. Even now, the number of responses from the Netherlands is quite small compared to the other countries. We are writing another article, now including the Netherlands, but exploring only other variables (age for learning how to cycle in different countries – and in fact children learn earlier in the Netherlands than in almost all of the other countries). Maybe the reason we have so few responses from the Netherlands is that they learn so early and it is such a common thing. In Portugal most parents recall when their kids learn how to cycle, but we had many people from the Netherlands saying they could not recall it.

2) And on a similar note: [9] recognises that the foundational movement skills are dependant on the context (ie location). looking at the responses. Roughly 25% of responses are from Latin America, 50% are from southern Europe and 25% from Northern Europe. Are the trends similar with in these regions? Or are they different?

Thank you for the suggestion. We did a new data analysis using the UN geoscheme classification to group the European countries in: “Southern Europe” and “Northern and Western Europe” and we considered Brazil and Mexico as “Latin America”. The trends seem to be similar in all these regions. We added this information in the revised Ms (please see discussion section). Regarding the brothers, there are differences between the 3 regions, the younger child learns significantly earlier than the older child in southern Europe and than the only child in northern Europe. There were no significant differences in Latin America. We also added this information in the revised Ms (please see discussion section).

3) Similarly in Northern Europe where there is a significant difference in what can be done outside during winter and summer (especially in Finland), did you take the time of year that the children were born into account? The time of year at which it is possible to learn to ride a bicycle could possibly be just as important as the factors you have investigated. For example learning on ice/snow might be a big cause for learning to cycle half a year later... 

Thank you for your comment. We did not. We believe that could be an interesting analysis if the main focus of the paper was the learning age and the differences in the learning age between regions, especially for northern countries. However, our aims were to investigate whether the age of learning to ride a bicycle was related with child’s frequency of physical activity and/or birth order.

4) I do not understand the numbers mentioned in the bars of figure 1 and 2. for example in figure one in the most left bar (less than 2x/month) it states 7.5+-5.3 . does this mean that the mean age was 7.5 with a standard deviation of 5.3 (ie 2.2years minimum to 12.8 maximum?) then the most right bar (Daily) comes to 3.5 minimum. ie significantly older than the first group... 

Yes, you are right. The numbers refer to the SD. We changed the figure caption to “Figure 1. Mean age and standard deviation to learn how to cycle by the frequency of physical activity practice (error bars represent 95% CI).” As you noted the SD is quite big in some cases but we believe that the 95% CI bars help to get the idea of where are the significant differences.

5) the following are a couple of details: 

line 129: states 10 languages, it is not clear if there are 2 "Portuguese" languages or just one, as Portugal and Brazil are mentioned, and another 9 languages.  

Thank you for noticing. We corrected line 129 to 9 languages. In fact, we were considering the 2 versions of Portuguese and counting them separately, but they are really the same language.

line 130: Finish = Finnish

Done.

line 230: thought = through ? 

Done.

Round 2

Reviewer 2 Report

Dear Authors, 

Thankyou for the much improved (in my opinion) manuscript and your comments. I have one remaining clarification about lines 129-132. "

Finally, the survey was translated into different languages being now available in 9 languages (Portuguese – from Portugal and Brazil, English, German, Croatian, Finnish, French, Dutch, Italian, Japanese and Spanish)."

1: Portuguese
2: English
3: German
4: Croatian
5: Finnish
6: French
7:  Dutch
8: Italian
9: Japanese
10: Spanish

is it thus not 10 languages? (my original question was is it 10 or 11? (depending on how you deal with Portuguese) so I'm suprized by the 9. unless you mean it was translated to another 9 languages. but then you dont need to mention Portuguese I suppose. 

As for our discussion on the Netherlands: the "loopfiets" (walking bicycle) is very popular here. most children learn to ride a bicycle on a loopfiets first. It becomes also a vague timespan at which they therefore can cycle. because they often start on a loopfiets, then later get a proper bicycle, but its not like a hard switch from loopfiets to bicycle, they often have them side by side, depending on the situation which one they will use, so there is a transition period as they grow to big for the loopfiets. the loopfiets initially is their favourite because its lighter and easier for them to re-orientate (as they can touch the ground with both feet). So at which point can they cycle? usually when they can roll down a hill on the loopfiets without their feet touching the ground.. but its not on a "bicycle" so its a vague situation. There are children that get their first loopfiets when they are not even a year old. initially they play with them inside the house. it can even be a way they learn to walk (extra leg/foot on the ground so to say)...  And something you do not touch on in this work is when do you constitute being able to cycle: is that with or without trainer wheels - I assume you mean without trainer wheels. but if you come from a loopfiets the only thing that the child still has to learn is the pedeaing motion. the balance has been mastered. going back to trainer wheels actually in that sense makes their life far more difficult as they have to learn a different control program! 

Author Response

Thankyou for the much improved (in my opinion) manuscript and your comments. I have one remaining clarification about lines 129-132. "

Finally, the survey was translated into different languages being now available in 9 languages (Portuguese – from Portugal and Brazil, English, German, Croatian, Finnish, French, Dutch, Italian, Japanese and Spanish)."

1: Portuguese
2: English
3: German
4: Croatian
5: Finnish
6: French
7:  Dutch
8: Italian
9: Japanese
10: Spanish

is it thus not 10 languages? (my original question was is it 10 or 11? (depending on how you deal with Portuguese) so I'm suprized by the 9. unless you mean it was translated to another 9 languages. but then you don’t’ need to mention Portuguese I suppose. 

Thank you for your comment. The previous comments and suggestions were very important for us to improve the Ms. You are absolutely right, the survey has been translated into 9 more languages but is available in 10, which created the previous confusion. We have changed this back in the text to 10 languages.

As for our discussion on the Netherlands: the "loopfiets" (walking bicycle) is very popular here. most children learn to ride a bicycle on a loopfiets first. It becomes also a vague timespan at which they therefore can cycle. because they often start on a loopfiets, then later get a proper bicycle, but its not like a hard switch from loopfiets to bicycle, they often have them side by side, depending on the situation which one they will use, so there is a transition period as they grow to big for the loopfiets. the loopfiets initially is their favourite because its lighter and easier for them to re-orientate (as they can touch the ground with both feet). So at which point can they cycle? usually when they can roll down a hill on the loopfiets without their feet touching the ground.. but its not on a "bicycle" so its a vague situation. There are children that get their first loopfiets when they are not even a year old. initially they play with them inside the house. it can even be a way they learn to walk (extra leg/foot on the ground so to say)...  And something you do not touch on in this work is when do you constitute being able to cycle: is that with or without trainer wheels - I assume you mean without trainer wheels. but if you come from a loopfiets the only thing that the child still has to learn is the pedeaing motion. the balance has been mastered. going back to trainer wheels actually in that sense makes their life far more difficult as they have to learn a different control program! 

Your comment is extremely enriching! In the first survey’s version we only asked "How old where you when you learned how to ride a bicycle?", but after applying the test version and collecting feedback we realised that this question was dubious. Participants did not understand that we mentioned a traditional bicycle (with pedals and without side training wheels) without help. So, in the final survey’s version we reworded this question to make it clearer and avoid doubts "How old where you when you first learned how to ride a bicycle independently and without training wheels?".

In this Ms we did not mentioned the influence of the training bicycle type, as the balance bike (loopfiets) or the bicycle with the training wheels because it wasn´t our purpose in specific Ms. Indeed, we have already published a preprint about it, you can see it in:

https://www.researchsquare.com/article/rs-268365/v1

And yes, you are completely right. The children who learned how to cycle with the balance bike and then transit to the traditional bicycle learned earlier than all the others. So, we also believed that the balance bike (loopfiets) can be the best learning instrument. 

Thank you for all your suggestions and discussion.